# Single Nucleotide Variants of the Human TIM-1 IgV Domain with Reduced Ability to Promote Viral Entry into Cells

**DOI:** 10.3390/v14102124

**Published:** 2022-09-26

**Authors:** Takanari Hattori, Takeshi Saito, Hiroko Miyamoto, Masahiro Kajihara, Manabu Igarashi, Ayato Takada

**Affiliations:** 1Division of Global Epidemiology, International Institute for Zoonosis Control, Hokkaido University, Sapporo 001-0020, Japan; 2International Collaboration Unit, International Institute for Zoonosis Control, Hokkaido University, Sapporo 001-0020, Japan; 3Department of Disease Control, School of Veterinary Medicine, University of Zambia, Lusaka 10101, Zambia; 4One Health Research Center, Hokkaido University, Sapporo 001-0020, Japan

**Keywords:** TIM-1, polymorphism, pseudotyped virus, glycoprotein, entry

## Abstract

Human T-cell immunoglobulin mucin 1 (hTIM-1) is known to promote cellular entry of enveloped viruses. Previous studies suggested that the polymorphisms of hTIM-1 affected its function. Here, we analyzed single nucleotide variants (SNVs) of hTIM-1 to determine their ability to promote cellular entry of viruses using pseudotyped vesicular stomatitis Indiana virus (VSIV). We obtained hTIM-1 sequences from a public database (Ensembl genome browser) and identified 35 missense SNVs in 3 loops of the hTIM-1 immunoglobulin variable (IgV) domain, which had been reported to interact with the Ebola virus glycoprotein (GP) and phosphatidylserine (PS) in the viral envelope. HEK293T cells transiently expressing wildtype hTIM-1 or its SNV mutants were infected with VSIVs pseudotyped with filovirus or arenavirus GPs, and their infectivities were compared. Eleven of the thirty-five SNV substitutions reduced the efficiency of hTIM-1-mediated entry of pseudotyped VSIVs. These SNV substitutions were found not only around the PS-binding pocket but also in other regions of the molecule. Taken together, our findings suggest that some SNVs of the hTIM-1 IgV domain have impaired ability to interact with PS and/or viral GPs in the viral envelope, which may affect the hTIM-1 function to promote viral entry into cells.

## 1. Introduction

The members of the T-cell immunoglobulin mucin (TIM) family are type Itransmembrane glycoproteins [1]. Of these, TIM-1 is expressed on various immune cells and a broad range of mucosal epithelial cells and plays roles in regulating immune responses, allergic responses, asthma, and transplant tolerance [2,3,4,5]. One of the physiological functions of TIM-1 is to recognize phosphatidylserine (PS) exposed on the apoptotic cell surface and mediate phagocytosis of those cells [6]. PS receptors such as TIM-1 are also known to recognize PS exposed on the viral envelope, and TIM-1 promotes viral entry into host cells through interactions with viral envelope-associated PS, which is known as “the viral apoptotic mimicry strategy” [7]. Mainly through this mechanism, TIM-1 is known to enhance the cellular entry of a wide range of enveloped viruses, including filovirus, flavivirus, alphavirus, arenavirus, rhabdovirus, and baculovirus [8,9]. In addition, previous studies demonstrated that the virus–TIM-1 interaction induced a release of inflammatory cytokines from primary CD4+ T cells and might also contribute to the pathogenesis of Ebola virus (EBOV) infection in mice [10,11]. TIM-1 was shown to activate dengue virus-induced autophagy through the TIM-1 signaling pathway [12]. Therefore, TIM-1 is thought to play an important role not only as a viral attachment receptor but also as an immune-regulatory molecule against some viruses.

TIM-1 is composed of an immunoglobulin variable (IgV) domain, mucin-like domain (MLD), transmembrane domain, and cytoplasmic tail (CT) (Figure 1a). The IgV domain is essential for the TIM-1-mediated enhancement of viral infections, and MLD is thought to be necessary to place the IgV domain within the appropriate distance from the cell surface [13,14]. Previous studies reported that genetic polymorphisms of human TIM-1 (hTIM-1) MLD and IgV domain affected its function to promote viral infections. A polymorphism with a six amino acid insertion into hTIM-1 MLD was associated with reduced and enhanced disease progression with human immunodeficiency virus and hepatitis A virus infection, respectively [15,16]. It was also reported that this insertion influenced in vitro cell susceptibility to the Japanese encephalitis virus [17]. A single nucleotide variant (SNV), the S51L variant (NCBI ID: rs2270922), of the hTIM-1 IgV domain, was reported to have reduced ability for the hTIM-1-mediated entry of pseudotyped lentivirus with chikungunya virus glycoprotein (GP) [18]. In addition, our previous study showed that a polymorphism of the TIM-1 IgV domain among African green monkey cell lines affected the efficiency of the cellular entry of vesicular stomatitis Indiana virus (VSIV) pseudotyped with EBOV GP [19]. From these findings, we hypothesized that SNVs of the hTIM-1 IgV domain might influence cell susceptibilities to viral infections. However, there are few studies focusing on the effects of SNVs of the TIM-1 IgV domain on viral entry into cells, and the information supporting this hypothesis is still limited.

The crystal structure of the IgV domain revealed that this domain has two antiparallel β sheets: a “BED-β sheet” consisting of β-strands B, E, and D and a “GFC-β sheet” consisting of β-strands A, G, F, C, C’, and C″ [20]. Three loops between β-strands B and C, C and C’, and F and G are named the BC loop, CC’ loop, and FG loop, respectively (Figure 1b,c). Six cysteine residues (C36, C46, C52, C57, C104, and C105) in the IgV domain form disulfide bonds on the GFC β-sheet. Two cysteine residues, C36 and C105, form a disulfide bond (C36-C105) between β-strand F and the BC loop, and four cysteine residues (C46, C52, C57, and C104) form two additional disulfide bonds (C46-C57 and C52-C104) that fix the CC’ loop onto the GFC β-sheet [21,22]. These bonds define a cleft formed by the CC’ and FG loops, providing a unique groove-like structure to bind PS, identified as the “PS-binding pocket” (Figure 1c). This structure is conserved among the TIM family and is also called the metal-ion-dependent ligand binding site (MILIBS), which was identified as an important region for binding to viral PS [21]. It has also been shown that the three loops (BC, CC’, and FG loops) of the IgV domain interact with both the EBOV GP and PS exposed on the viral envelope [23]. These findings led us to hypothesize that SNV substitutions in the IgV domain might affect the function of hTIM-1 to promote viral entry into cells.

In this study, the potential effects of hTIM-1 SNV substitutions on cellular entry of enveloped viruses were evaluated using pseudotyped VSIVs. We focused on SNVs of the IgV domain as a key part for binding to virus particles and found 35 missense SNVs in the public gene database (Ensembl genome browser). Of these, some SNV substitutions reduced the efficiency of hTIM-1-mediated cellular entry of VSIVs having GPs of different origins. The present study suggests that some SNVs of the hTIM-1 IgV domain may have a lower ability to bind PS and/or viral GPs on pseudotyped VSIV particles.

## 2. Materials and Methods

### 2.1. SNV Information on the hTIM-1 IgV Domain

Nucleotide sequences of the hTIM-1 IgV domain were obtained from the Ensembl genome browser [24] and the Single Nucleotide Polymorphism Database (dbSNP) on the National Center for Biotechnology Information (NCBI) website [25], and 35 missense SNVs were found in the IgV domain. The NCBI ID numbers of these SNVs are as follows: rs774628607 (S31F), rs766684661 (V32F), rs763305471 (T33A), rs748542797 (L34P), rs1235087840 (P35S), rs948562287 (Y38H), rs1331131690 (T43A), rs1392286629 (S44P), rs201914430 (M45V), rs778524415 (C46W), rs368474218 (W47R), rs1334308674 (R49S), rs1467830805 (G50S), rs2270922 (S51L), rs766596791 (S53F), rs750683624 (C57S), rs1324375875 (Q58R), rs765450007 (G60D), s776921169 (V62I), rs1313131093 (G101D), rs759044943 (V102L), rs770585374 (R106H), rs748923252 (V107I), rs769720430 (H109P), rs370980439 (R110C), rs1196575610 (G111R), rs1240319173 (F113L), rs377678930 (N114S), rs1168125347 (D115G), rs745941787 (M116L), rs778900665 (T119I), rs754029647 (V120I), rs1190295106 (S121P), rs1169032336 (I124T), rs556857102 (V125L) (Table 1).

Allele frequencies of hTIM-1 SNVs were obtained from the Genome Aggregation Database (genomAD) and Trans-Omics for Precision Medicine (TOPMed). The potential detrimental effects of hTIM-1 SNV substitutions on its structure and function were assessed by sorting intolerant from tolerant (SIFT), polymorphism phenotyping 2 (PolyPhen-2), combined annotation dependent depletion (CADD), and rare exome variant ensemble learner (REVEL). The SIFT scores are classified as tolerated (T) (≥0.05) or deleterious (D) (<0.05). PolyPhen-2 scores are classified as benign (B) (<0.5), possibly damaging (PosD) (0.5–0.9), or probably damaging (ProD) (≥0.9). CADD scores are classified as likely benign (LB) (<20) or likely deleterious (LD) (≥20). REVEL scores are classified as likely benign (LB) (<0.5) or likely disease-causing (LDC) (≥0.5). The scoring methods were taken from the following Ensemble genome browser [26].

### 2.2. Cells

Human adenocarcinoma-derived alveolar basal epithelial A549, human embryonic kidney (HEK) 293T, and African green monkey kidney epithelial Vero E6 cells were grown in Dulbecco’s Modified Eagle’s Medium (DMEM) (Sigma-Aldrich, St. Louis, MO, USA) supplemented with 10% fatal calf serum (FCS) (Sigma-Aldrich, St. Louis, MO, USA), 100 U/mL penicillin, and 0.1 mg/mL streptomycin (Gibco, Waltham, MA, USA) at 37 °C in a 5% CO_2_ incubator. Expi 293F cells (Gibco, Waltham, MA, USA) were maintained in Expi 293F Expression Medium (Thermo Fisher Scientific, Waltham, MA, USA) as described in the manufacturer’s instructions and incubated at 37 °C in an 8% CO_2_ incubator while shaking at 125 rpm.

### 2.3. hTIM-1 Plasmids

Total RNA was extracted from A549 cells using Trizol (Invitrogen, Carlsbad, CA, USA), and then a cDNA library was prepared with SuperScriptTM IV (Invitrogen, Carlsbad, CA, USA) and the oligo dT20 primer (5′-TTTTTTTTTTTTTTTTTTTT-3′). To amplify the coding region of the hTIM-1 gene, polymerase chain reaction (PCR) was performed with a KOD One polymerase (TOYOBO, Osaka, Japan) using the primers EcoRI-hTIM-1-F (5′-ATAGAATTCGCCACCATGCATCCTCAAGTG-3′) containing an EcoRI restriction site and XhoI-hTIM-1-R (5′-TATCTCGAGCTATTATTCCAAGCGGCTTCG-3′) containing an XhoI restriction site. After sequence confirmation of the wildtype (WT) hTIM-1 gene (GenBank accession number NM_012206.3), this PCR product was inserted into an expression plasmid, pCAGGS. Using this plasmid as a template, a plasmid encoding 6 × histidine-tagged WT soluble hTIM-1 (aa 1-295) was generated by PCR-based mutagenesis. SNV substitutions were produced by site-directed mutagenesis using a KOD One polymerase (TOYOBO, Osaka, Japan) with primers containing the desired nucleotide substitutions. All mutations were confirmed by Sanger sequencing of the plasmids.

### 2.4. Generation of Pseudotyped VSIVs

Filovirus and arenavirus GPs were used in the present study. Ebola virus (EBOV), Marburg virus (MARV), and Lloviu virus (LLOV) were selected from 3 filovirus genera, Ebolavirus, Marburgvirus, and Cuevavirus, respectively [27]. Two arenaviruses, Junin virus (JUNV) and Lassa virus (LASV), were selected as representatives of the New World and Old World arenaviruses, respectively [28]. The GPs of these enveloped viruses are the only proteins that are responsible for virus entry into host cells [29]. Using replication-incompetent VSIV containing the green fluorescent protein (GFP) gene instead of the VSIV G protein gene (VSIVΔG*-VSIV), VSIVs bearing GPs of EBOV (Mayinga), MARV (Angola), LLOV (Asturias), JUNV (Candid #1), and LASV (Josiah) were generated as described previously [30], and designated VSIVΔG*-EBOV, -MARV, -LLOV, -JUNV, and -LASV, respectively. Briefly, HEK293T cells were transfected with the pCAGGS plasmid encoding each viral GP gene. After 24-h incubation, the culture medium was removed and the cells were incubated with VSIVΔG*-VSIV at a multiplicity of infection of 2.0 for 60 min at 37 °C. After the inoculum was removed and the cells were washed 3 times with DMEM, the medium was replaced with fresh DMEM with 10% FCS. Sixteen hours later, the culture supernatants were harvested and stored at −80 °C until use. Virus infectious units (IUs) in HEK293T cells were determined by counting the number of GFP-positive cells with an IN Cell Analyzer 2500 (GE Healthcare, Little Chalfont, UK). Previous studies reported that hTIM-1 enhanced the infectivities of envelope viruses, including VSIV, and viral attachment through the TIM-1-PS interaction has been proposed as one of the general mechanisms for the first step of viral entry into cells [7]. In addition, the TIM-1-PS interaction was shown to be important for the infectivities of pseudotyped VSIVs by the competition assay with PS liposomes, and the presence of PS on the VSIV envelope was demonstrated in the binding assay using Annexin V [14].

### 2.5. Virus Entry Assay

HEK293T cells lacking endogenous expression of hTIM-1 [5] were seeded in 96-well plates (1.0 × 10^4^ cells per well) precoated with poly-L-lysine (Cultrex, R&D Systems, Minneapolis, MN, USA). Twenty-four hours later, the cells were transfected with 0.2 μg/well of pCAGGS encoding WT hTIM-1, its SNV mutants, or an empty plasmid as a negative control. At 24 h post-transfection, these cells were infected with VSIVΔG*-EBOV, -MARV, -LLOV, -JUNV, -LASV, or -VSIV. Each virus was appropriately diluted to provide 200–500 IUs/well in HEK293T cells and then incubated with an anti-VSIV G protein monoclonal antibody VSIV-G (N) 1-9 for 30 min at 37 °C to abolish the background infectivity of parental VSIVΔG*-VSIV (i.e., inoculum virus) [30]. Twenty-four hours later, IUs were determined by counting the numbers of GFP-expressing cells using an IN Cell Analyzer 2500 (GE Healthcare, Little Chalfont, UK). The relative infectivity was determined by setting the IU value of empty plasmid-transfected cells to 100%.

### 2.6. Sodium Dodecyl Sulfate-Polyacrylamide Gel Electrophoresis (SDS-PAGE) and Western Blotting

HEK293T were seeded in 24-well plates (1.0 × 10^4^ cells per well) precoated with poly-L-Lysine (Cultrex, R&D Systems, Minneapolis, MN, USA). After overnight culture, the cells were transfected with the plasmids encoding WT hTIM-1, its SNV mutants, or the empty plasmid as a negative control. After 48-h incubation, these cells were washed with phosphate-buffered saline (PBS) 3 times and lysed with a radioimmunoprecipitation assay (RIPA) buffer (0.25 mM EDTA, pH 8.0, 25 mM Tris/HCl, pH 7.6, 150 mM NaCl, 1% NP-40, 1% sodium deoxycholate, 0.1% SDS). After centrifugation, the supernatants were mixed with SDS-PAGE sample buffer (Bio-Rad, Hercules, CA, USA) with 5% 2-mercaptoethanol and subjected to 10% SDS-PAGE. Separated proteins were blotted on a polyvinylidene difluoride membrane (Merck Millipore Corporation, Burlington, MA, USA). After blocking with 5% skim milk, the membrane was incubated with a goat anti-hTIM-1 polyclonal antibody (AF1750, R&D Systems, Minneapolis, MN, USA) and anti-β-actin mouse monoclonal antibody (AC15, Abcam, Cambridge, UK) as primary antibodies for 1 h, respectively. After washing with PBS containing 0.05% Tween 20 (PBST), the membrane was incubated with a horseradish peroxidase (HRP)-conjugated donkey anti-goat IgG polyclonal antibody (705-035-003, Jackson ImmunoResearch, West Grove, PA, USA) and goat anti-mouse IgG polyclonal antibody (115-035-062, Jackson ImmunoResearch, West Grove, PA, USA) as secondary antibodies for 1 h, respectively. After washing with PBST, the bound antibodies were visualized with Immobilon Western (Merck Millipore Corporation, Burlington, MA, USA). The relative expression levels (i.e., the band intensities of hTIM-1 molecules divided by those of β-actin) were analyzed using an Amersham Imager 600 (GE Healthcare, Little Chalfont, UK).

### 2.7. Immunofluorescence Assay

HEK293T cells were seeded in a μ-Slide 8-Well Chamber Slide-well (iBidi GmbH, Martinsried, Germany) after precoating with poly-L-Lysine (Cultrex, R&D Systems, Minneapolis, MN, USA). After overnight culture, the cells were transfected with the plasmids encoding WT hTIM-1 or its SNV mutant genes. At 24 h post-transfection, the cells were washed with PBS and fixed with 4% paraformaldehyde in PBS for 15 min. After washing with PBS, the cells were incubated with PBS containing 3% bovine serum albumin for blocking for 1 h at room temperature. After washing 3 times with PBST, the cells were incubated with a goat anti-hTIM-1 polyclonal antibody (AF1750, R&D Systems, Minneapolis, MN, USA) as the primary antibody for 1 h at room temperature. The cells were washed 3 times with PBST and then incubated with a donkey anti-goat IgG polyclonal antibody conjugated with fluorescein isothiocyanate (FITC) (sc2024, Santa Cruz Biotechnology, Santa Cruz, CA, USA) as a secondary antibody, followed by counterstaining with 1 μg/mL 4′,6-diamidino-2-phenylindole, dihydrochloride (DAPI) (D1306, Molecular Probes, Eugene, OR, USA) for 1 h in the dark at 4 °C. Images were acquired with a 63 × oil objective lens on a Zeiss LSM700 inverted microscope using ZEN 2009 software (Carl Zeiss, Oberkochen, Germany).

### 2.8. Flow Cytometry

HEK293T cells were seeded in 6-well plates (2.5 × 10^4^ cells per well) precoated with poly-L-lysine (Cultrex, R&D Systems, Minneapolis, MN, USA) and incubated overnight at 37 °C. The cells were transfected with pCAGGS encoding WT hTIM-1 or its SNV mutant genes. At 24 h post-transfection, the cells were washed with PBS, detached by treatment with 0.25% trypsin, and then fixed with 4% paraformaldehyde in PBS for 15 min. After washing with PBS, the cells were incubated with a goat anti-hTIM-1 polyclonal antibody (AF1750, R&D Systems, Minneapolis, MN, USA) for 1 h at room temperature. Then the cells were washed with PBST and stained with a donkey anti-goat IgG polyclonal antibody conjugated with FITC (sc2024, Santa Cruz Biotechnology, Santa Cruz, CA, USA) for 30 min at 4 °C in the dark. After washing 3 times with PBST, the percentage of FITC-positive cells and mean fluorescent intensity (MFI) of FITC signals were analyzed using a FACSCanto flow cytometer (BD Biosciences, San Jose, CA, USA) and FlowJo software (Tree Star, San Carlos, CA, USA).

### 2.9. Purification of Soluble hTIM-1 Proteins

To produce soluble hTIM-1 proteins, Expi 293F cells (Gibco, Waltham, MA, USA) were transfected with the plasmids encoding 6 × histidine-tagged WT or 11 SNV mutants of soluble hTIM-1. The cells were cultured for 4 days and the supernatants were collected and filtered with a 0.45 μm pore membrane (Sartorius Stedim, Goettingen, Germany). Using the Ni-NTA purification system (Invitrogen, Carlsbad, CA, USA), soluble hTIM-1 proteins were purified from the supernatants and concentrated with Amicon Ultra 50K (Merck Millipore, Darmstadt, Germany). The purified proteins were analyzed in SDS-PAGE for their purity and then stored at −30 °C until use.

### 2.10. Viral Entry Inhibition Assay Using Soluble hTIM-1 Proteins

Vero E6 cells were seeded in 96-well plates (3.0 × 10^4^ cells per well). After overnight incubation, equal volumes of VSIVΔG*-EBOV (2000–3000 IUs/well) diluted in DMEM with 2% FCS and 20 μg/mL (final concentration) of purified soluble hTIM-1 proteins were mixed and incubated for 30 min at room temperature, and then added to Vero E6 cells in 5% CO_2_ at 37 °C. IgG S139/1 (a monoclonal antibody specific to Influenza A virus hemagglutinin) was used as a negative control [31]. Twenty-four hours later, IUs were determined by counting the numbers of GFP-expressing cells with an IN Cell Analyzer 2500 (GE Healthcare, Little Chalfont, UK). The relative percentage of infectivity was determined by setting the IU value of cells infected with the virus alone to 100%.

### 2.11. Statistical Analysis

All statistical analyses were performed using R software (Version 3.6.0). For the comparison of relative infectivities and expression levels of hTIM-1, one-way analysis of variance followed by the Dunnett test was used. *p*-values of less than 0.05 were considered statistically significant.

## 3. Results

### 3.1. SNVs of the hTIM-1 IgV Domain

We focused on the 3 loops (BC, CC’, and FG) and connecting β-strands in the hTIM-1 IgV domain, which have been shown to be important for binding to EBOV GP and PS [23]. The BC, CC’, and FG loops (positions 35–44, 50–59, and 101–125, respectively) are adjacent to 5 β-strands (B, C, C’, F, and G at positions 31–34, 45–49, 60–64, 101–107, and 116–125, respectively) (Figure 1b,c). First, amino acid positions at which SNVs had been reported were extracted (Figure 1b). In total, 51 missense SNVs in 35 positions were found in this region. When several different amino acid residues were reported at one SNV position, we selected the residue with the highest allele frequency of SNVs and examined 35 missense SNVs in the following experiments: 4 SNVs (S31F, V32F, T33A, and L34P) in β-strand B, 4 SNVs (P35S, Y38H, T43A, and S44P) in the BC loop, 4 SNVs (M45V, C46W, W47R, and R49S) in β-strand C, 5 SNVs (G50S, S51L, S53F, C57S, and Q58R) in the CC’ loop, 2 SNVs (G60D and V62I) in β-strand C’, 4 SNVs (G101D, V102L, R106H, and V107I) in β-strand F, 6 SNVs (H109P, R110C, G111R, F113L, N114S, and D115G) in the FG loop, and 6 SNVs (M116L, T119I, V120I, S121P, I124T, and V125L) in β-strand G (Figure 1b,c and Table 1).

### 3.2. hTIM-1 SNV Substitutions Affecting the Entry of Pseudotyped VSIV into HEK293T Cells

We then evaluated the effects of hTIM-1 SNV substitutions on the infectivities of pseudotyped VSIVs using HEK293T cells lacking the hTIM-1 expression on the surface [5]. HEK293T cells transfected with hTIM-1-expressing plasmids were infected with VSIVΔG*-EBOV, -MARV, -LLOV, -JUNV, -LASV, or -VSIV, and relative infectivities were compared to those of the cells expressing WT hTIM-1 (Figure 2). As expected, the expression of WT hTIM-1 enhanced the cellular entry of VSIVΔG*-EBOV, -LLOV, and -JUNV, whereas weaker enhancement was seen for VSIVΔG*-MARV and -VSIV. Consistent with the previous report, VSIVΔG*-LASV infection was not enhanced by the hTIM-1 expression in HEK293T cells [9]. These results indicated that the efficiency of hTIM-1-mediated viral entry varied depending on viral surface GPs even in the same pseudotyping condition (i.e., VSIV). While none of the SNV substitutions significantly affected the infectivity of VSIVΔG*-LASV, 7 SNV substitutions (L34P, C46W, W47R, C57S, H109P, N114S, and D115G) uniformly reduced the ability of hTIM-1 to promote entry of the other pseudotyped viruses and VSIVΔG*-VSIV. On the other hand, the I124T substitution affected pseudotyped viruses (VSIVΔG*-EBOV, -MARV, -LLOV, and -JUNV) but not VSIVΔG*-VSIV. Two SNV substitutions (R110C and G111R) significantly reduced the ability of hTIM-1 against VSIVΔG*-EBOV, -LLOV, -JUNV, and -VSIV, but limited effects were observed against VSIVΔG*-MARV. In contrast, the V62I substitution only affected VSIVΔG*-JUNV. Taken together, these results indicated that most of the SNV substitutions that reduced the ability of hTIM-1-mediated viral entry were common among these viruses but that there might be a slightly different manner of action depending on the virus.

To evaluate the expression levels of exogenously introduced hTIM-1 in HEK293T cells, the band intensities were compared among WT hTIM-1 and its SNV mutant proteins in SDS-PAGE and Western blot analysis (Figure 3a). There was no statistically significant difference in hTIM-1 expression levels compared to the value of the WT hTIM-1 molecule. We next confirmed the intracellular localization and cell surface expression levels of hTIM-1 proteins in HEK293T cells using immunofluorescence assay and flow cytometry. As expected, endogenous hTIM-1 expression was not detected in HEK293T cells, and SNV substitutions that significantly reduced the hTIM-1-mediated viral entry had no significant effect on cell surface localization of hTIM-1 proteins (Figure 3b). The cell surface expression levels were further assessed by flow cytometry (Figure 3c,d). For the percentages of FITC-positive cells and the MFI, there was no significant difference compared to WT hTIM-1 (Figure 3c,d).

### 3.3. SNV Substitutions Affecting Neutralizing Activity of Soluble hTIM-1 against VSIVΔG*-EBOV

hTIM-1 expressed on the cell membrane is cleaved by a matrix metalloproteinase upstream of the transmembrane domain, resulting in the production of the soluble form of hTIM-1 [32,33]. Soluble forms of hTIM-1 are reported to inhibit some viral infections and are expected to be therapeutic candidates [13,15,34,35,36]. To further assess the effects of SNV substitutions on viral entry, purified soluble hTIM-1 proteins containing each representative SNV substitution were produced and their neutralizing activities against VSIVΔG*-EBOV were compared (Figure 4). As expected, the soluble form of WT hTIM-1 significantly inhibited viral infection by about 50%, whereas no inhibitory effect was seen in a negative control (CTR IgG). Soluble hTIM-1 proteins with the V62I substitution also significantly inhibited infection as efficiently as WT hTIM-1, suggesting that this substitution did not significantly affect the hTIM-1 function consistent with the data of cellular entry assay for VSIVΔG*-EBOV (Figure 2). On the other hand, the other 10 SNVs tested for soluble forms, all of which significantly reduced the infectivity of VSIVΔG*-EBOV as described above (see Figure 2), showed less ability to inhibit VSIVΔG*-EBOV infection than WT and V62I hTIM-1. Indeed, no significant differences compared to CTR IgG were detected in 8 of the 10 SNV mutants, and only slight inhibitory activity was detected in 2 SNV substitutions (N114S and I124T).

### 3.4. Mapping of the hTIM-1 SNV Positions

In total, 11 SNV substitutions in the hTIM-1 IgV domain (L34P, C46W, W47R, C57S, V62I, H109P, R110C, G111R, N114S, D115G, and I124T) were found to reduce the hTIM-1-mediated cellular entry of the pseudotyped VSIVs (Figure 2). Structural analyses revealed that the amino acid residues at positions 57, 109, 110, 111, 114, 115, and 124 were exposed on the surface of the IgV domain, whereas the amino acid residues at positions 34, 46, 47, and 62 were not located on the molecular surface (Figure 5). Of these, 5 SNV positions (109, 110, 111, 114, and 115) were located around the PS-binding site (MILBS motif, 112–115) in the FG loop. On the other hand, 6 SNV positions (34, 46, 47, 57, 62, and 124) were located far from this site. Position 34 was located on the boundary between β-strand B and the BC loop, positions 46 and 47 were located on β-strand B, and position 57 was located on the CC’ loop. Two cysteine residues at positions 46 and 57 forming a disulfide bond bridged the CC’ loop and the GFC β-sheet [21]. Positions 62 and 124 were located on β-strands C’ and G, respectively.

### 3.5. Allele Frequencies and Functional Predictions of hTIM-1 SNVs

We found that the allele frequencies of most of the SNVs examined in this study were very low in the global population (Table 2 and 3). The W47R variant with the highest minor allele frequency (MAF) among the SNVs was also found to be a rare variant (MAF = 1.44 × 10^−4^). Then, potential modifications of the hTIM-1 structure caused by the SNV substitutions were predicted using bioinformatics tools, including SIFT, PolyPhen-2, CADD, and REVEL from the Ensembl genome browser (Table 2). The V62I substitution was predicted to have no effect on the protein structure, whereas L34P, C46W, W47R, C57S, and D115G SNV substitutions were predicted to disrupt the functional structure by all the bioinformatics tools used in this study. G111R, N114S, and I124T substitutions were predicted to have some negative effects by SIFT, PolyPhen-2, and CADD, whereas H109P and R110C substitutions were predicted to damage the hTIM-1 structure only by SIFT and PolyPhen-2. In summary, among the 11 SNV substitutions that reduced the TIM-1-mediated viral entry, 10 substitutions were predicted to likely have the potential to alter the hTIM-1 function. In contrast, SNV substitutions that had no significant effect on viral entry showed a tendency to be predicted to cause a mild or no change in its function (Table 3).

## 4. Discussion

It has been reported that polymorphisms of hTIM-1 are associated with the pathogenesis of atopic dermatitis, allergy, rheumatoid arthritis, asthma, systemic lupus erythematosus, and viral infections [37,38,39]. In general, genetic polymorphisms of a host protein have the potential to influence its structure and function and have been shown to affect the susceptibility of hosts and/or severity of viral infectious diseases [40,41,42,43]. Since hTIM-1 is one of the major attachment receptors for some viruses, substitutions in hTIM-1 SNVs might influence the susceptibility of humans to viral infections. However, the information on hTIM-1 SNVs affecting virus infectivity is still limited. Here, we focused on 35 missense SNV substitutions located in the hTIM-1 IgV domain and found that some SNVs might have reduced ability to promote virus infections.

The PS-binding activity of the IgV domain is thought to be important for the attachment of some enveloped viruses, and the PS-binding site was identified as MILIBS consisting of 4 amino acid residues (WFND motif: W112, F113, N114, and D115) in the FG loop [21,22]. Accordingly, N114D and D115A substitutions in this motif were shown to abolish flavivirus infection [44]. Consistent with this previous finding, 5 SNV substitutions, N114S, and D115G, as well as H109P, R110C, and G111R, N114S, around the PS-binding site in the FG loop influenced the cellular entry of the pseudotyped VSIV (Figure 2 and Figure 5c). These SNV substitutions may directly alter the PS-binding site, resulting in the reduced ability of hTIM-1 to interact with PS on the viral envelope.

Five SNV substitutions (L34P, C46W, W47R, C57S, and V62I) between β-strands B and C’, which are not directly involved in the PS-binding pocket, also reduced the viral entry into cells. In addition, 4 of these amino acids (i.e., positions 34, 46, 47, and 62) are not even exposed on the surface of the hTIM-1 molecule. It is assumed that these SNV substitutions might potentially cause structural distortions of the IgV domain, leading to reduced interaction with PS and/or viral GPs. For example, the L34P substitution might affect the flexibility of the BC loop since this amino acid position is located at the root of the BC loop inside the molecule (Figure 5a). The C46W and C57S substitutions might also alter the conformation of the molecule since these amino acid substitutions abolish the disulfide bond (C46-C57) between β-strand C and the FG loop, which plays an important role in maintaining the PS-binding pocket [21]. We previously reported that single amino acid substitutions at position 48 in African green monkey-derived TIM-1 affected the infectivity of VSIVs pseudotyped with filovirus GPs [19]. Interestingly, this position is close to the 2 SNV substitutions (C46W and W47R in β-strand C), resulting in the reduction of the virus infectivity in the present study, suggesting that these consecutive amino acid residues (i.e., positions 46–48) are important for the hTIM-1 function as an attachment receptor for viral infection.

Interestingly, the I124T substitution in β-strand G connecting to the FG loop reduced the viral entry. Although this amino acid position is located far from the PS-binding site, the I124T substitution might affect the stability of the FG loop, indirectly reducing the PS-binding activity. It might also be possible that the I124T substitution reduced the interaction with EBOV, MARV, LLOV, and JUNV GPs since the infectivity of VSIVΔG*-VSIV was only limitedly affected by this substitution. It was also noted that the V62I substitution significantly reduced the cellular entry of VSIVΔG*-JUNV but not the other viruses. It is also noted that the S51L substitution reduced the hTIM-1-mediated enhancement of pseudotyped lentivirus bearing chikungunya virus GP [18], but this substitution resulted in no significant change in the VSIV pseudotyped with filovirus and arenavirus GP in the present study. A previous study demonstrated that hTIM-1 binds to the receptor binding domain of EBOV GP in addition to PS, although the interaction site between EBOV GP and hTIM-1 is still elusive [23]. Importantly, our findings may also suggest the mechanisms underlying the binding of hTIM-1 to viral GPs. Crystal structures of the complex of GPs and hTIM-1 will be needed to determine its molecular basis.

In another study that investigated single amino acid substitutions in the hTIM-1 IgV domain using an alanine scanning method, 8 substitutions (Y38A, F55A, R106A, G111A, F113A, N114A, D115A, and K117A located on the BC, CC’ and FG loops) were found to decrease the efficiency of GP-mediated EBOV entry into cells [14]. However, no significant change in the VSIVΔG*-EBOV infectivity was found in the hTIM-1 SNV mutants with Y38H and R106H substitutions in the present study, while 2 substitutions (H109A and R110A located on the FG loop) that affected the virus infectivity in the present study did not decrease the GP-mediated EBOV entry into cells in a previous study [14]. Such differences suggested that the polarity of amino acids at these positions may be important for the interaction between hTIM-1 and PS and/or viral GPs.

The present study suggests that hTIM-1 polymorphisms may affect cell susceptibility to viral infections. Eleven SNV substitutions were found to reduce the cellular entry of VSIV pseudotyped viruses, but it is indeed still unclear which process (i.e., viral attachment, internalization, or membrane fusion) was affected by these substitutions. Although these SNV substitutions are thought to reduce viral attachment, further studies are needed to clarify the mechanisms of the SNV-associated dysfunction of hTIM-1. Another limitation of this study is that pseudotyped VSIVs were used, and the hTIM-1-mediated cellular entry might be largely dependent on the VSIV envelope. We assume that biological properties, such as the lipid composition of the envelope and density of viral glycoproteins on the viral particle, are different between pseudotyped VSIVs and authentic filoviruses/arenaviruses. To further investigate the association of genetic polymorphisms in hTIM-1 and viral infections, both in vitro and in vivo experiments using the authentic viruses are needed in the future. It would also be of interest to investigate the hTIM-1 SNVs for other physiological roles such as immune-regulatory function.

## Figures and Tables

**Figure 1 viruses-14-02124-f001:**
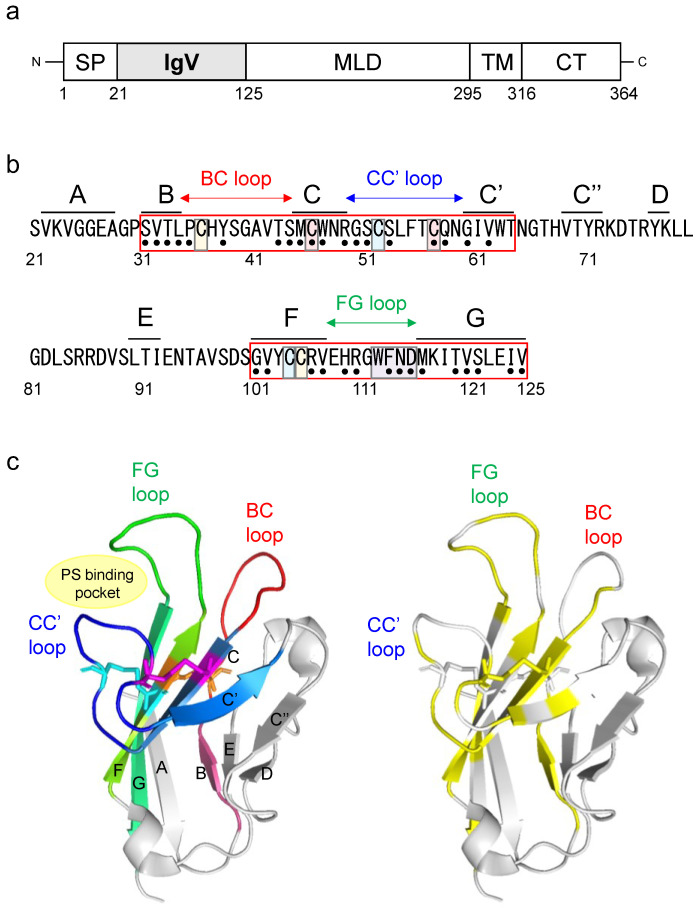
Structure of the hTIM-1 IgV domain and the 35 SNV substitutions examined in this study. (**a**) Schematic diagram of hTIM-1. TIM-1 consists of a signal peptide (SP), immunoglobulin variable (IgV) domain, mucin-like domain (MLD), transmembrane domain (TM)5, and cytoplasmic tail (CT). (**b**) Amino acid sequence of the hTIM-1 IgV domain (positions 21–125; GenBank accession number: NM_012206.3). Regions corresponding to each β-strand are shown above the sequence. Six cysteine residues forming three disulfide bonds (C36-C105, C46-C57, and C52-C104) are highlighted in orange, magenta, and cyan pairs, respectively. The PS-binding site (MILIBS-WFND motif) is highlighted in purple. Black dots under the sequence indicate the 35 SNV positions examined in this study. (**c**) Crystal structure of the hTIM-1 IgV domain (PDB ID: 5DZO). In the left panel, β-sheets are labeled with their corresponding β-strand names, and five β-strands, B, C, C’, F, and G, are colored in warm pink, sky blue, marine, chartreuse, and lime green, respectively. BC, CC’, and FG loops are colored in red, blue, and green, respectively. Amino acid residues involved in three disulfide bonds (C36-C105, C46-C57, and C52-C104) are shown in orange, magenta, and cyan, respectively. In the right panel, the 35 SNV positions are shown in yellow.

**Figure 2 viruses-14-02124-f002:**
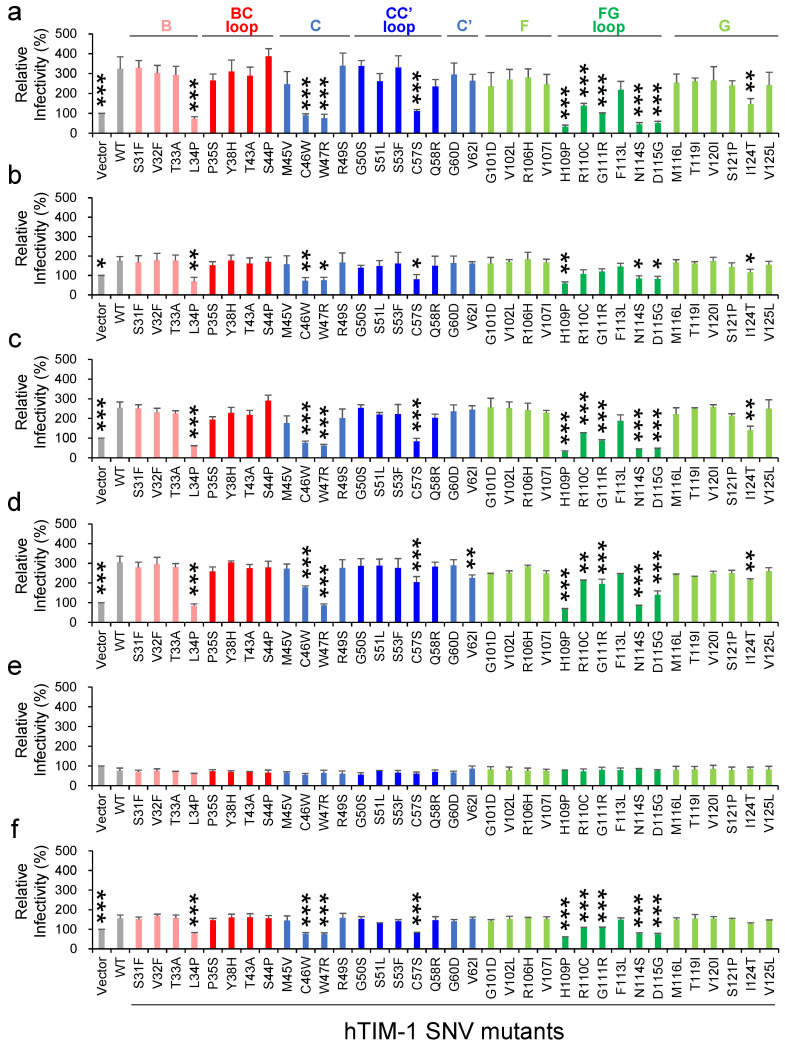
Effects of 35 SNV substitutions in the hTIM-1 IgV domain on the cellular entry of pseudotyped VSIVs. HEK293T cells transfected with the plasmids encoding WT hTIM-1, its SNV mutants, or the empty plasmid were infected with VSIVΔG*-EBOV (**a**), -MARV (**b**), -LLOV (**c**), -JUNV (**d**), -LASV (**e**), or -VSIV (**f**). Twenty-four hours later, infectious units (IUs) were determined by counting the numbers of GFP-positive cells and the relative infectivities were determined by setting the IU value of empty plasmid-transfected cells to 100%. The means and standard deviations (SDs) of 3 independent experiments are shown. Statistical significance was calculated compared to WT using the Dunnett test (* *p* < 0.05, ** *p* < 0.01, *** *p* < 0.001).

**Figure 3 viruses-14-02124-f003:**
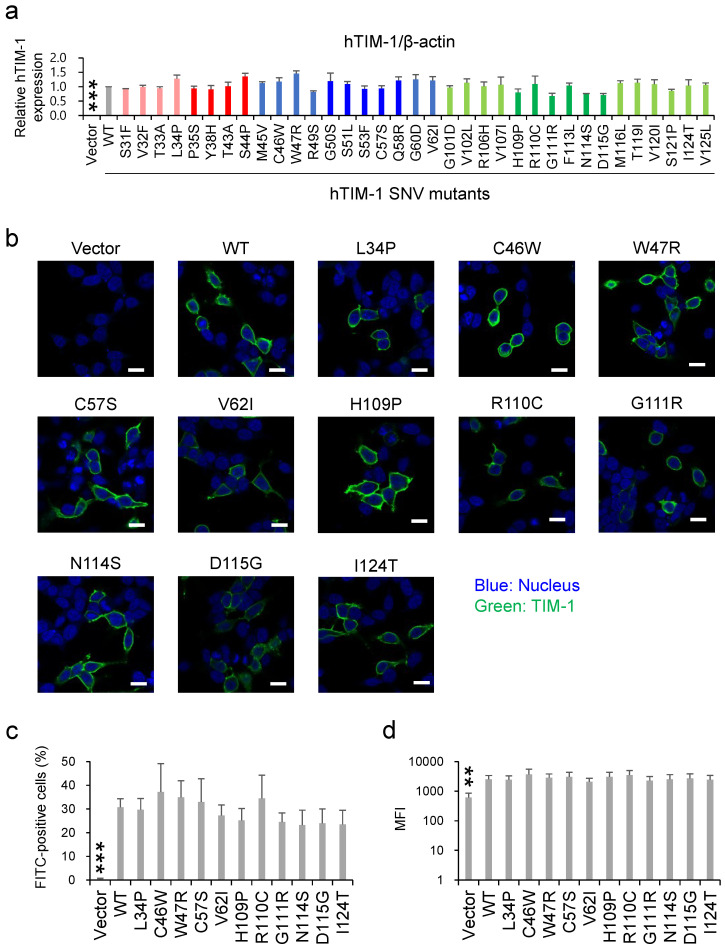
Expression levels of exogenous hTIM-1 proteins in HEK293T cells. (**a**) HEK293T cells transfected with the plasmids encoding WT hTIM-1, its SNV mutants, or the empty plasmid were harvested at 48 h post-transfection and analyzed in SDS-PAGE and Western blotting. The amounts of β-actin in the total cell lysates were also analyzed as an internal control. The relative expression level was determined as the ratio of the band intensity of hTIM-1 to β-actin, and that of each SNV mutant was compared with that of WT hTIM-1. The means and SDs of 3 independent experiments are shown. Statistical significance was calculated compared to WT using the Dunnett test (*** *p* < 0.001). (**b**) HEK293T cells transfected with the plasmids encoding WT hTIM-1, its SNV mutants, or empty plasmid were immunostained as described in the Materials and Methods section. The cell images were captured with confocal microscopy. The scale bars represent 10 μm. (**c**,**d**) HEK293T cells transfected with the plasmids and immunostained as described above were also analyzed with a FACSCanto flow cytometer. The cell surface expression of hTIM-1 was quantified as percentages of the FITC-positive cells (**c**) and mean fluorescent intensity (MFI) of the FITC signals (**d**). Statistical significance was calculated compared to the WT using the Dunnett test (** *p* < 0.01, *** *p* < 0.001).

**Figure 4 viruses-14-02124-f004:**
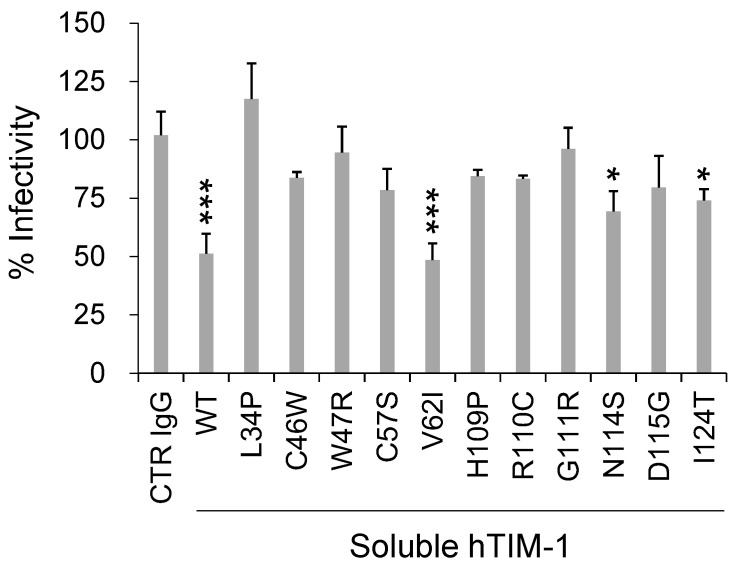
Effects of SNV substitutions on the ability of soluble hTIM-1 to inhibit cellular entry of VSIVΔG*-EBOV into Vero E6 cells. Vero E6 cells were infected with VSIVΔG*-EBOV preincubated with 20 μg/mL soluble forms of WT hTIM-1 or its SNVs mutants for 30 min room temperature. An anti-Influenza A virus hemagglutinin monoclonal antibody was used as a negative control (CTR IgG). After 24-h incubation, GFP-positive cells were counted. The relative percentage of infectivity was calculated by setting the IU value of cells infected with virus alone to 100%. Each bar represents the means and SDs of triplicate wells. Statistical significance was calculated compared to CTR IgG using the Dunnett test (* *p* < 0.05, *** *p* < 0.001).

**Figure 5 viruses-14-02124-f005:**
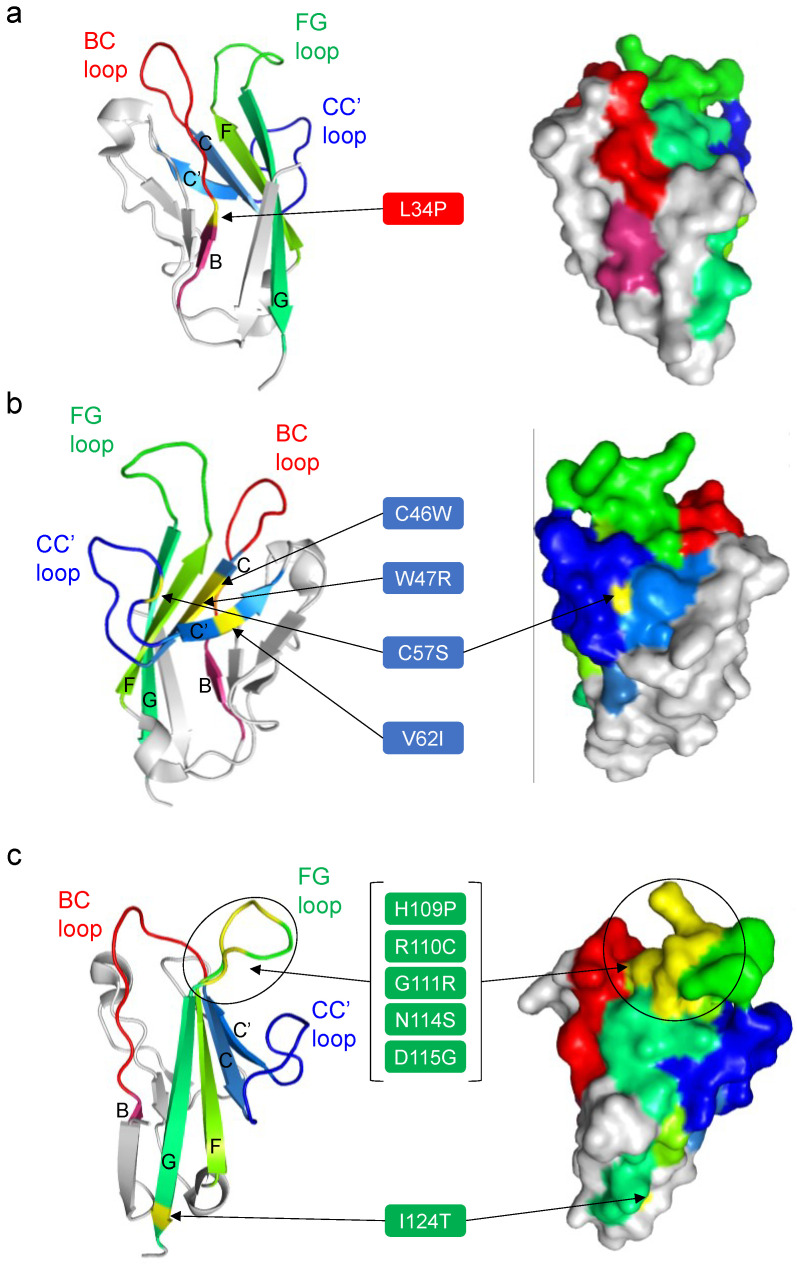
Amino acid positions of the hTIM-1 SNV substitutions that significantly reduced the ability to promote the cellular entry of the pseudotyped viruses. The amino acid positions of the 11 SNV substitutions that reduced the cellular entry of VSIV-based pseudotyped viruses in the present study are shown in yellow in the crystal structure of the IgV domain in the hTIM-1 molecule (PDB ID: 5DZO). Other coloring is the same as in Figure 1. To focus on these amino acid positions and locations of the BC, CC’, and FG loops, the 11 SNVs were separately mapped in 3 figures; BC loop (**a**), CC’ loop (**b**), and FG loop (**c**). Ribbon (left panel) and surface (right panel) representations of the hTIM-1 IgV domain are shown. In the left panels, 5 β-strands (B, C, C’, F, and G) are indicated with their corresponding letters.

**Table 1 viruses-14-02124-t001:** Thirty-five SNVs of hTIM-1 IgV domain listed in the public database.

rsID ^a^	Position	Amino Acid Wildtype/Mutant	SNV Name	Location ^b^
rs774628607	31	S/F	S31F	β-strand B
rs766684661	32	V/F	V32F	β-strand B
rs763305471	33	T/A	T33A	β-strand B
rs748542797	34	L/P	L34P	β-strand B
rs1235087840	35	P/S	P35S	BC loop
rs948562287	38	Y/H	Y38H	BC loop
rs1331131690	43	T/A	T43A	BC loop
rs1392286629	44	S/P	S44P	BC loop
rs201914430	45	M/V	M45V	β-strand C
rs778524415	46	C/W	C46W	β-strand C
rs368474218	47	W/R	W47R	β-strand C
rs1334308674	49	R/S	R49S	β-strand C
rs1467830805	50	G/S	G50S	CC’ loop
rs2270922	51	S/L	S51L	CC’ loop
rs766596791	53	S/F	S53F	CC’ loop
rs750683624	57	C/S	C57S	CC’ loop
rs1324375875	58	Q/R	Q58R	CC’ loop
rs765450007	60	G/D	G60D	β-strand C’
s776921169	62	V/I	V62I	β-strand C’
rs1313131093	101	G/D	G101D	β-strand F
rs759044943	102	V/L	V102L	β-strand F
rs770585374	106	R/H	R106H	β-strand F
rs748923252	107	V/I	V107I	β-strand F
rs769720430	109	H/P	H109P	FG loop
rs370980439	110	R/C	R110C	FG loop
rs1196575610	111	G/R	G111R	FG loop
rs1240319173	113	F/L	F113L	FG loop
rs377678930	114	N/S	N114S	FG loop
rs1168125347	115	D/G	D115G	FG loop
rs745941787	116	M/L	M116L	β-strand G
rs778900665	119	T/I	T119I	β-strand G
rs754029647	120	V/I	V120I	β-strand G
rs1190295106	121	S/P	S121P	β-strand G
rs1169032336	124	I/T	I124T	β-strand G
rs556857102	125	V/L	V125L	β-strand G

^a^ NCBI ID numbers of hTIM-1 SNVs. ^b^ The amino acid positions were identified from a crystal structure of the hTIM-1 IgV domain.

**Table 2 viruses-14-02124-t002:** Information of hTIM-1 SNV substitutions that significantly reduced the cellular entry of pseudotyped viruses.

SNVs	Global MAF ^a^	SIFT ^b^	PolyPhen-2 ^c^	CADD ^d^	REVEL ^e^
L34P	7.39 × 10^−5^	D	ProD	LD	LDC
C46W	3.94 × 10^−5^	D	ProD	LD	LDC
W47R	1.44 × 10^−4^	D	ProD	LD	LDC
C57S	6.57 × 10^−6^	D	ProD	LD	LDC
V62I	3.94 × 10^−5^	T	B	LB	LB
H109P	4.02 × 10^−6^	D	PosD	LB	LB
R110C	1.97 × 10^−5^	D	PosD	LB	LB
G111R	6.57 × 10^−6^	D	ProD	LD	LB
N114S	4.09 × 10^−6^	D	ProD	LD	LB
D115G	2.63 × 10^−5^	D	ProD	LD	LDC
I124T	7.96 × 10^−6^	D	ProD	LD	LB

^a^ Global minor allele frequencies (MAFs) of hTIM-1 SNVs were obtained from gnomAD and TOPMed. ^b^ SIFT: Sorting intolerant from tolerant, D: Deleterious, T: Tolerated. ^c^ PolyPhen-2: Polymorphism Phenotyping-2, B: Benign, PosD: Possibly damaging, ProD: Probably damaging. ^d^ CADD: Combined annotation dependent depletion, LB: Likely benign, LD: Likely deleterious. ^e^ REVEL: Rare exome variant ensemble learner, LB: Likely benign, LDC: Likely disease-causing.

**Table 3 viruses-14-02124-t003:** Information of hTIM-1 SNV substitutions that did not significantly affect the cellular entry of pseudotyped viruses.

SNVs	Global MAF ^a^	SIFT ^b^	PolyPhen-2 ^c^	CADD ^d^	REVEL ^e^
S31F	NA ^f^	D	PosD	LB	LB
V32F	NA	D	ProD	LB	LB
T33A	6.57 × 10^−6^	D	PosD	LB	LB
P35S	6.57 × 10^−6^	T	PosD	LB	LB
Y38H	NA	D	ProD	LD	LB
T43A	4.01 × 10^−6^	D	B	LB	LB
S44P	4.01 × 10^−6^	D	B	LB	LB
M45V	1.00 × 10^−4^	D	PosD	LB	LB
R49S	6.57 × 10^−6^	D	ProD	LB	LB
G50S	4.01 × 10^−6^	D	ProD	LB	LB
S51L	3.28 × 10^−5^	D	B	LD	LB
S53F	4.00 × 10^−6^	D	PosD	LD	LB
Q58R	4.01 × 10^−6^	T	B	LB	LB
G60D	2.00 × 10^−5^	T	B	LB	LB
G101D	4.00 × 10^−6^	D	PosD	LD	LDC
V102L	3.29 × 10^−5^	T	B	LB	LB
R106H	3.29 × 10^−5^	D	ProD	LD	LDC
V107I	NA	T	B	LB	LB
F113L	2.63 × 10^−5^	D	ProD	LD	LB
M116L	1.25 × 10^−5^	T	B	LB	LB
T119I	4.20 × 10^−6^	D	PosD	LB	LB
V120I	4.60 × 10^−5^	T	B	LB	LB
S121P	6.57 × 10^−6^	D	ProD	LB	LB
V125L	1.97 × 10^−5^	D	B	LB	LB

^a–e^ The abbreviations and criteria are described in the footnote of Table 2. ^f^ NA: Information is not available in the databases.

## Data Availability

The data presented in this study are available on request from the corresponding author.

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
