# Peer review of "Single Nucleotide Variants of the Human TIM-1 IgV Domain with Reduced Ability to Promote Viral Entry into Cells"

_viruses, 2022, doi:10.3390/v14102124_

Round 1
Reviewer 1 Report
The manuscript “Single nucleotide variants of the human TIM-1 IgV domain 2 with reduced ability to promote viral entry into cells” by Hattori et al. reports the analyses of the single nucleotide polymorphisms of the gene coding for the TIM-1 and their effect on the cell susceptibility to viral infection. In particular, using a recombinant VSV pseudotyped with the envelope of filovirus and arenavirus. They show that polymorphisms can affect the efficiency of viral infection and speculates on the effect of the polymorphisms on the TIM-1 structure and interaction with the viral particles.
The manuscript is well written and the experiments well done. Results support the authors’ conclusions but considering some comments of the discussion paragraph the manuscript could significantly increase its impact adding few experiments (with the main relevant polymorphisms and at least EBOV) to better characterize the role of the polymorphisms on viral attachment. Experiments of viral attachment (see for example Luganini et al Front Microbiol. 2018 Aug 7;9:1826 or Mercorelli et al Antimicrob Agents Chemother. 2010 Nov;54(11):4561-7.) could be useful to obtain strongest results.
Author Response
The manuscript is well written and the experiments well done. Results support the authors’ conclusions but considering some comments of the discussion paragraph the manuscript could significantly increase its impact adding few experiments (with the main relevant polymorphisms and at least EBOV) to better characterize the role of the polymorphisms on viral attachment. Experiments of viral attachment (see for example Luganini et al Front Microbiol. 2018 Aug 7;9:1826 or Mercorelli et al Antimicrob Agents Chemother. 2010 Nov;54(11):4561-7.) could be useful to obtain strongest results.
Response: Thank you for the suggestion. We conducted the viral attachment assay according to the recommended references. Briefly, HEK293T cells transiently expressing WT hTIM-1, its SNV mutants, or the empty gene were prechilled for 10 min at 4°C and infected with VSIVΔG*-EBOV for 4 hours at 4°C. After washing with PBS to remove the unattached virus, cells were replaced with fresh 10% FCS-DMEM and incubated at 37°C. Twenty-four hours later, GFP-positive cells were counted. However, against our expectations, WT hTIM-1 expression could not enhance the cellular entry of VSIVΔ* EBOV in our assay. A previous study also conducted the viral attachment assay using HEK293T cells permanently expressing TIM-1 and replication-competent recombinant VSIV pseutotyped with EBOV GP, and they showed the TIM-1-mediated enhancement (Microbiol Spectr. 2022 Jun ;10(3):e0221221.). Although it is unclear why our assay did not work, we assumed that transiently expressed TIM-1 from the pCAGGS plasmid might not be sufficient to enhance the attachment during the 4-hour incubation period under the cool condition. Although it was difficult to evaluate the details of the viral entry process in the present study, it is important to clarify the role of SNVs in the viral entry process as the next step. Thus, we added the following sentence in Discussion:
“Eleven SNV substitutions were found to reduce the cellular entry of VSIV pseudotyped viruses, but it is indeed still unclear which process (i.e., viral attachment, internalization, or membrane fusion) were affected by these substitutions. Although these SNV substitutions are thought to reduce viral attachment, further studies are needed to clarify the mechanisms of the SNV-associated disfunction of hTIM-1” (Lines 490-494)
Reviewer 2 Report
The authors have conducted a comprehensive study on effect of single nucleotide variants (SNVs) of hTIM-1 on viral entry. While the study is well designed, and a lot of work has been put into the manuscript there are a few major critiques (please see below):
Major comments:
1. The central premise of authors’ argument is that hTIM-1 is a phosphatidyl serine (PS) receptor. They argue that PS receptors such as TIM-1 also recognize PS exposed on the viral envelope, and promote viral through interactions with viral envelope-associated PS. The authors have however not presented any data on the presence of PS in their pseudotyped viruses. On the same lines there are more questions (see minor comments).
2. Pseudotyping systems often have variable efficiencies of GP incorporation. Amount of GP present on the surface can affect entry efficiency. While authors explore the effect of variable hTIM-1 levels on virus entry, they do not present any such data about the pseudotypes. Showing the relative amount of GP incorporation in the pseudotype will therefore be useful.
3. Authors have presented structural data about different hTIM-1 SNVs. They have however not modeled the viral glycoproteins (particularly EBOV) on this and how these changes may affect glycoprotein binding? It will be immensely informative to have that data.
Minor comments:
1. Is the amount of PS on different pseudotypes the same?
2. How does changing PS incorporation affect entry?
3. How is this data different from the previous studies that authors have themselves cited?
Author Response
Major comments:
- The central premise of authors’ argument is that hTIM-1 is a phosphatidylserine (PS) receptor. They argue that PS receptors such as TIM-1 also recognize PS exposed on the viral envelope, and promote viral through interactions with viral envelope-associated PS. The authors have however not presented any data on the presence of PS in their pseudotyped viruses. On the same lines there are more questions (see minor comments).
Response: Thank you for your comment. Several previous studies reported that TIM-1 enhanced the infectivities of enveloped viruses including VSIV, and viral attachment through the TIM-1-PS interaction has been proposed as one of the general mechanisms for the first step of viral entry into cells (Nat Rev Microbiol. 2015 Aug;13(8):461-9). In addition, the TIM-1-PS interaction was shown to be important for the infectivities of pseudotyped VSIVs by the competition assay with PS liposomes, and the presence of PS on the VSIV envelope was demonstrated in the binding assay using Annexin V (J Virol. 2013 Aug; 87(15): 8327-8341). Therefore, we assume that PS is exposed on the VSIV envelope. Thus, we added the following sentence in Materials and Methods:
“Previous studies reported that hTIM-1 enhanced the infectivities of envelope viruses including VSIV, and viral attachment through the TIM-1-PS interaction has been proposed as one of the general mechanisms for the first step of viral entry into cells [7]. In addition, the TIM-1-PS interaction was shown to be important for the infectivities of pseudotyped VSIVs by the competition assay with PS liposomes, and the presence of PS on the VSIV envelope was demonstrated in the binding assay using Annexin V [14]” (Lines 159-165)
- Pseudotyping systems often have variable efficiencies of GP incorporation. Amount of GP present on the surface can affect entry efficiency. While authors explore the effect of variable hTIM-1 levels on virus entry, they do not present any such data about the pseudotypes. Showing the relative amount of GP incorporation in the pseudotype will therefore be useful.
Response: Thank you for your comment. As you pointed out, the amount of GP on the VSIV envelope may affect the efficiency of the TIM-1-mediated entry. This limitation of the study is mentioned in the last paragraph of Discussion. However, it is indeed difficult to compare GP incorporation rates among pseudotyped viruses because there is no antibody that equally detect the glycoproteins of the different viruses used in this study. In addition, the purpose of this study is to investigate whether hTIM-1 SNV substitutions could affect the function to promote viral entry but not to compare hTIM-1-mediated enhancement among viruses. We believe that the impact of hTIM-1 SNVs on the infectivity was properly assessed.
- Authors have presented structural data about different hTIM-1 SNVs. They have however not modeled the viral glycoproteins (particularly EBOV) on this and how these changes may affect glycoprotein binding? It will be immensely informative to have that data.
Response: Thank you for your comment. A previous study demonstrated the binding affinity between hTIM-1 IgV mutants and EBOV GP using surface plasmon resonance assay and 3 loops were identified as the regions important for the binding between TIM-1 and EBOV GP (Protein Cell. 2015, 6, 814–824). However, since the crystal structure of the complex of EBOV GP and hTIM-1 is still unavailable, we could not analyze the interface between TIM-1 and EBOV GP. We added the following sentence in discussion:
“Crystal structures of the complex of GPs and hTIM-1 will be needed to determine its molecular basis” (Lines 477-478)
Minor comments:
- Is the amount of PS on different pseudotypes the same?
Response: Thank you for your comment. In the present study, the amount of PS expressed on the viral surface might be largely dependent on the property of VSIV envelope. However, the expression of each viral GP may also affect the biological properties such as the lipid composition of the envelope, the actual amount of PS among viruses could not be verified in the present study. As noted in the last paragraph in Discussion (Lines 494-499), we consider these facts as the limitation of our study. Again, since the purpose of this study is not to compare hTIM-1-mediated enhancement among viruses, the potential difference in the PS amount is not the major issue in the present study.
- How does changing PS incorporation affect entry?
Response: Thank you for your comment. It may be interesting to see how changing PS incorporation affect viral entry into cell. Although we think that increased PS amounts might result in enhanced entry mediated by hTIM-1, it seems difficult to show its evidence. However, as mentioned above, comparison of the PS amount in the viral envelope is not within the scope of the present study.
- How is this data different from the previous studies that authors have themselves cited?
Response: Thank you for your comment. Indeed, there are few studies focusing on the effects of SNVs of the TIM-1 IgV domain on viral entry into cells, and the information supporting this hypothesis is still limited. We added this statement (Lines 64-66). We cited Ref #14 (Moller-Tank, et al., J Virol 87, 8327–8341, 2013) in which alanine scanning of hTIM-1 was performed but they did not focus on SNV substitutions. Although some amino acid positions examined are similar but some different results were obtained most likely because of the polarity of substituted amino acids. Difference between their and our studies has been mentioned in Discussion (Lines 479-488). We cited Ref #18 (Kirui, et al., Cells 10, 1828, 2021) reporting that the TIM-1 SNV substitution (S51L) in the IgV domain reduced the cellular entry of pseudotyped lentivirus with chikungunya virus GP. However, this substitution resulted in no significant change in the VSIV pseudotyped with filovirus and arenavirus GP in the present study. We added this observation in Discussion (Line 471-474).
Following sentences were added:
“However, there are few studies focusing on the effects of SNVs of the TIM-1 IgV domain on viral entry into cells, and the information supporting this hypothesis is still limited” (Lines 64-66)
“It is also noted that the S51L substitution reduced the hTIM-1-mediated enhancement of pseudotyped lentivirus bearing chikungunya virus GP [18], but this substitution re-sulted in no significant change in the VSIV pseudotyped with filovirus and arenavirus GP in the present study” (Lines 471-474)
Round 2
Reviewer 1 Report
The modification of the manuscript is acceptable. It is a pity that the viral adhesion assay did not work in the experimental model used by Authors. However, the conclusion of the manuscript could be reasonable.
Minor: check the title of the paragraph 2.6: "2.6. Sodium dodecyl surface-polyacrylamide.... is the word surface correct?
Author Response
Reviewer #1
Minor: check the title of the paragraph 2.6: "2.6. Sodium dodecyl surface-polyacrylamide.... is the word surface correct?
Response. Thank you for this comment. We corrected to “sulfate”.
Reviewer 2 Report
No further comments.
Author Response
Reviewer #2
No further comments.
Response. Thank you very much.